# Ripply2 recruits proteasome complex for Tbx6 degradation to define segment border during murine somitogenesis

**Wei Zhao[1,2†‡], Masayuki Oginuma[1†§], Rieko Ajima[1,3,4†], Makoto Kiso[1,3], Akemi Okubo[1], Yumiko Saga[1,2,3,4]\***

[1]Division of Mammalian Development, National Institute of Genetics, Mishima, Japan; [2]Department of Biological Sciences, Graduate School of Science, The University of Tokyo, Tokyo, Japan; [3]Mouse Research Supporting Unit, National Institute of Genetics, Mishima, Japan; [4]Department of Genetics, SOKENDAI, Mishima, Japan

**\*For correspondence:**
ysaga@nig.ac.jp

[†]These authors contributed equally to this work

**Present address:** [‡]Laboratory for Human Organogenesis, RIKEN Center for Developmental Biology, Chuo-ku, Hyogo; [§]Laboratory of Integrated Signaling Systems, Department of Molecular Medicine, Institute for Molecular and Cellular Regulation, Gunma University, Maebashi, Japan

**Competing interests:** The authors declare that no competing interests exist.

**Abstract** The metameric structure in vertebrates is based on the periodic formation of somites from the anterior end of the presomitic mesoderm (PSM). The segmentation boundary is defined by the Tbx6 expression domain, whose anterior limit is determined by Tbx6 protein destabilization via Ripply2. However, the molecular mechanism of this process is poorly understood. Here, we show that Ripply2 directly binds to Tbx6 in cultured cells without changing the stability of Tbx6, indicating an unknown mechanism for Tbx6 degradation in vivo. We succeeded in reproducing in vivo events using a mouse ES induction system, in which Tbx6 degradation occurred via Ripply2. Mass spectrometry analysis of the PSM-fated ES cells revealed that proteasomes are major components of the Ripply2-binding complex, suggesting that recruitment of a protein-degradation-complex is a pivotal function of Ripply2. Finally, we identified a motif in the T-box, which is required for Tbx6 degradation independent of binding with Ripply2 in vivo.
DOI: https://doi.org/10.7554/eLife.33068.001

## Introduction

The transient expression of T-box transcriptional factors in limited groups of cells is a common process during embryogenesis for the determination and differentiation of special tissues such as in limb and heart development (*Papaioannou, 2014*; *Wilson and Conlon, 2002*). During somitogenesis, somites form periodically from the anterior end of the PSM toward the posterior direction every 2 hr, which coordinates a balance between maintenance of the PSM and differentiation of epithelial somites. The expression of Tbx6 starts from the progenitors of the posterior mesoderm and extends to the anterior part of presomitic mesoderm (PSM), then progressively regresses from the anterior edge of its expression domain when a new somite is formed. Tbx6 maintains the mesodermal properties of somite progenitor cells (*Chapman and Papaioannou, 1998*), but induces the expression of segmentation genes in the anterior PSM to establish the new somite boundary (*Oginuma et al., 2008*).

The temporal periodicity of somitogenesis is established in the posterior PSM via the function of a so-called molecular clock, which is operated by complex gene regulatory networks under the control of three major signaling pathways, Notch, FGF and Wnt (*Hubaud and Pourquié, 2014*). The periodicity of this segmentation clock is translated into the activation of the segmentation gene *Mesoderm posterior protein 2 (Mesp2)* around the segmental border (*Morimoto et al., 2005*). *Mesp2* expression is temporally regulated by Notch signaling, and spatially defined by Tbx6; both factors work positively and coordinate each other (*Yasuhiko et al., 2006*; *Yasuhiko et al., 2008*).

The anterior limit of the *Mesp2* mRNA expression domain is consistent with the Tbx6 anterior limit. Once translated, Mesp2 induces the expression of its target gene *Ripply2*, then Ripply2 suppresses Tbx6 protein, which results in the termination of *Mesp2* transcription (*Oginuma et al., 2008*; *Zhao et al., 2015*). This Tbx6-Mesp2-Ripply2 reciprocal regulation is the spatial mechanism that successively defines the position of the next anterior border of Mesp2, by which the metronomic segmented somites with determined size are correctly generated (*Morimoto et al., 2007*; *Takahashi et al., 2010*). The activation/inactivation switch for Tbx6 is also a typical behavior among T–box transcriptional factors, which play important roles in development during embryogenesis such as Tbx3 in ICM development (*Davenport et al., 2003*), Eomes in blastocytes (*Ciruna and Rossant, 1999*; *Strumpf et al., 2005*), and Tbx1, Tbx2, Eomes in limb development (*Hancock et al., 1999*).

The negative feedback loop of Ripply2-Tbx6 for the termination of Mesp2 activity during each somitic cycle is the fundamental process to create the spatial periodicity of the segmented somites in mice. Recently, both zebrafish ripply1/2 and mouse Ripply2 proteins were found to play a role in the degradation of T-box family factors (*Wanglar et al., 2014*; *Zhao et al., 2015*). Our previous study demonstrated that ectopic Ripply2 expression in the posterior PSM was sufficient for the destabilization of T-box factors- Tbx6 and T protein (*Zhao et al., 2015*). However, the molecular nature of Ripply2-mediated destabilization is poorly understood.

In this study, we found that Tbx6 and Ripply2 interacted with each other, but Tbx6 degradation never occurred in cultured cells, indicating that the PSM tissue is necessary for Tbx6 degradation. However, it is difficult to use PSM tissue from embryos for biochemical analyses because the population of Ripply2[+] cells in the PSM is very low (only approximately 1000 ~ 3000 cells/embryo, depending on somitic phases). Thus, we established an induction system for PSM-like cells (we refer to this system as the PSM-fated induction system) using mouse ES cells, by which we reproduced the Tbx6 expression/degradation in cultured cells. We used this system to search for factors interacting with Ripply2. We also used BAC-transgenic mice and chimera mice produced by CRISPR/Cas9 engineered *Tbx6*[Tbx6-venus] ES cells to examine the requirements of a motif in Tbx6 that is essential for degradation in vivo.

## Results

### Ripply2 directly interacted with Tbx6 but did not lead to destabilization of Tbx6 in cultured cells

Based on our previous study demonstrating that Ripply2 expression is sufficient for inducing Tbx6 destabilization in mouse PSM tissue (*Zhao et al., 2015*), we presumed that Ripply2 and Tbx6 interacted directly, as reported for Zebrafish and Xenopus (*Hitachi et al., 2009*; *Kawamura et al., 2008*). The presence of a so-called Ripply homology domain (amino acids FPIQ), implicated in interaction with the T-box domain (*Kawamura et al., 2008*), suggested that mouse Ripply2 also has the ability to bind Tbx6. To examine their interaction, we co-transfected Myc-Ripply2 and FLAG-Tbx6 expressing constructs into HEK293T cells. The potential interaction was analyzed by immunoprecipitation using anti-FLAG antibody followed by western blotting with anti-Myc antibody. A strong association between FLAG-Tbx6 with Myc-Ripply2 was observed with HEK293T cell lysates (*Figure 1A*). To further explore whether the association between these two proteins is direct, we carried out an in vitro GST pull-down assay using bacterially expressed GST-Tbx6 and His-Ripply2 fusion proteins. GST-Tbx6 fusion protein, but not GST alone, was able to pull down His-Ripply2, indicating that Tbx6 directly interacts with Ripply2 (*Figure 1B*).

Next, we asked whether the T-box is sufficient for Ripply2-binding by testing different constructs; one that expressed only the T-box domain (FLAG-T-box), one that expressed the C-terminal deleted Tbx6 (FLAG-1-T-box) and one that lacked the T-box domain (FLAG-Tbx6$_{\Delta T\text{-box}}$). We found that the T-box domain alone was sufficient for association with Ripply2 and that other regions of Tbx6 were dispensable for this interaction (*Figure 1C*). We also generated Ripply2 mutants to test the requirement of the FPIQ tetrapeptide motif implicated in T-box binding (Myc-Ripply2$_{\Delta FPIQ}$) and the WRPW motif, a known Groucho binding domain (Myc-Ripply2$_{\Delta WRPW}$) (*Hancock et al., 1999*). We found that both these mutants lacked the ability to bind Tbx6 (*Figure 1D*), indicating that these conserved sequences are required for Ripply2-mediated Tbx6 suppression. In zebrafish and *Xenopus*, ripply family factors repress mesp by suppressing the transcriptional activity of tbx6 as a transcriptional-

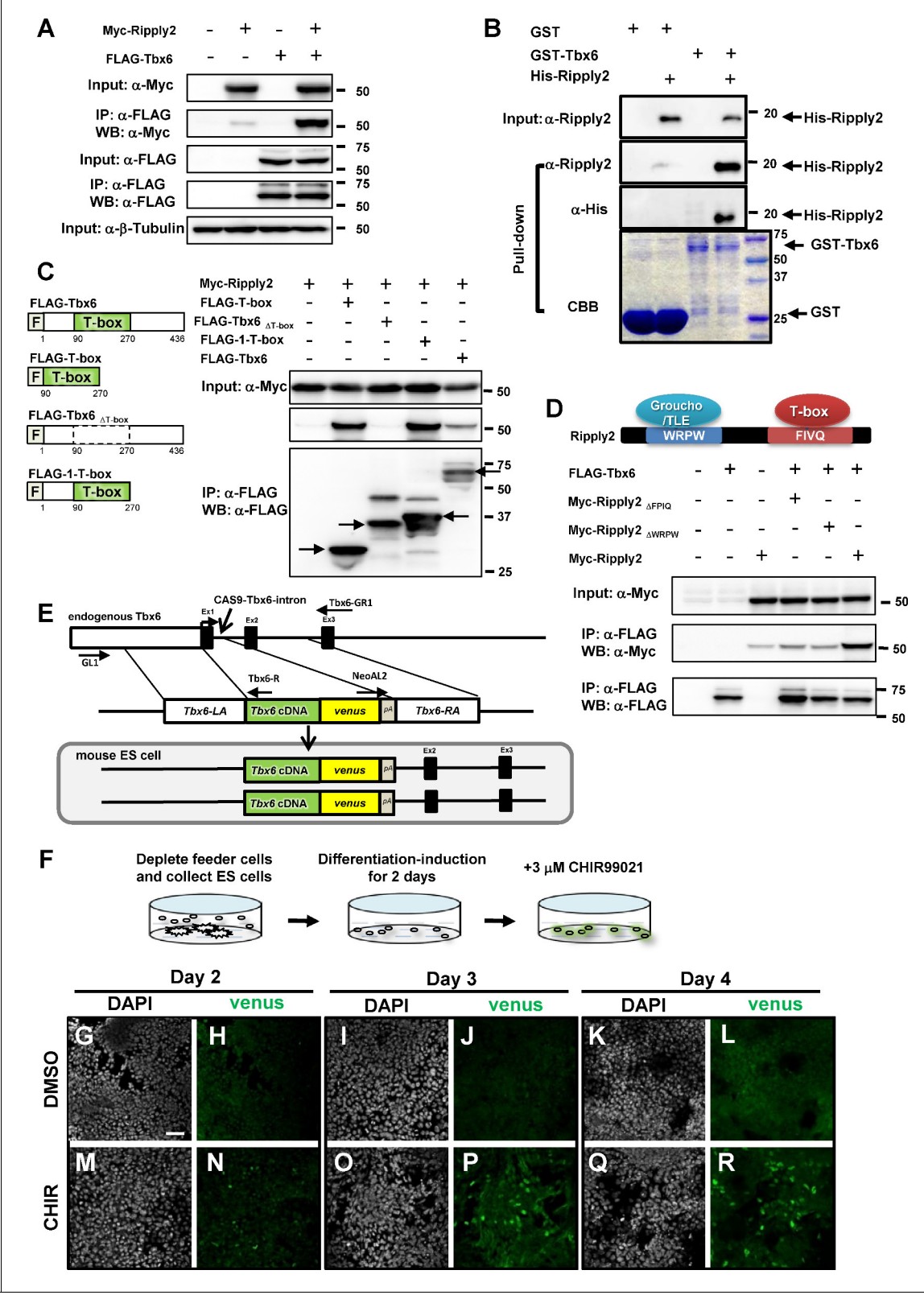

**Figure 1.** Establishment of the PSM-fated ES cell system. (**A**) Immunoprecipitation (IP) for examining Tbx6-Ripply2 interaction. DNA constructs for FLAG-Tbx6 was co-transfected with Myc-Ripply2 into HEK293T cells. IP was conducted using anti-FLAG beads, followed by western blotting for detecting FLAG-Tbx6 and Myc-Ripply2 by anti-FLAG and anti-Myc antibodies (N = 10). (**B**) GST pull-down assay to examine the direct interaction between Tbx6 and Ripply2. Purified His-Ripply2 protein was mixed with cell lysate of GST-Tbx6. Anti-Ripply2 antibody and anti-His antibody was used
*Figure 1 continued on next page*

*Figure 1 continued*

to detect His-Ripply2 signal (N = 2). (C) IP-western analyses to determine Tbx6 domain for Ripply2 interaction. DNA constructs for FLAG-Tbx6, FLAG-T-box, FLAG-Tbx6$_{\Delta T\text{-box}}$ or FLAG-1-T-box, were co-transfected with Myc-Ripply2 into HEK293T cells. IP was conducted using anti-FLAG beads, followed by western blotting for detecting FLAG-Tbx6 and Myc-Ripply2 by anti-FLAG and anti-Myc antibodies (N = 6). Arrows indicate protein bands showing expected molecular size. (D) IP for Tbx6 and mutant Ripply2. FLAG-Tbx6 was co-transfected with Myc-Ripply2$_{\Delta FPIQ}$, Myc-Ripply2$_{\Delta WRPW}$, or Myc-Ripply2 (wild-type) into the HEK293T cultured cells. Cell lysates were incubated with anti-FLAG beads. Western blotting was conducted using anti-FLAG and anti-Myc antibodies (N = 6). (E) Strategy for establishing the *Tbx6-venus* knock-in (KI) ES cell line. *Tbx6* cDNA connected with the venus sequence replaced exon-1 *via* Cas9-aided homologous recombination. (F) Schematic diagram for the method of PSM differentiation. (G–R) Time course change of Tbx6-venus protein expression in PSM-fated ES cells at 2 (G–N), 3 (I–P), or 4 (K–R) days after the addition of either DMSO or CHIR99021 in culture medium. The Tbx6-venus signals were detected by anti-GFP antibody. (N = 2) Scale bar: 50 μm.

DOI: https://doi.org/10.7554/eLife.33068.002

The following figure supplements are available for figure 1:

**Figure supplement 1.** Luciferase reporter assay showing no effect of mouse Ripply2 and TLE1 on mouse Tbx6 transcriptional activity.
DOI: https://doi.org/10.7554/eLife.33068.003

**Figure supplement 2.** Ectopic expression of Ripply2 in the heart had no influence on the heart development.
DOI: https://doi.org/10.7554/eLife.33068.004

repressor together with the Groucho co-repressor (*Kawamura et al., 2005*; *Kawamura et al., 2008*; *Kondow et al., 2007*). However, the transcriptional activity of mouse Tbx6 was not influenced by mouse Ripply2 and TLE1 (homolog of Groucho in mouse) (*Figure 1—figure supplement 1*).

Despite Tbx6-Ripply2 interaction, Tbx6 was never destabilized in HEK293T cells even in the presence of Ripply2 (*Figure 1A*). In addition, the ectopic expression of Ripply2 in the cardiac mesoderm did not influence on the stability of T-box factors Tbx18 and Tbx5 in the heart (*Figure 1—figure supplement 2*), which is different from the destabilizing effect of Ripply2 on Tbx6 and Brachyury (T) in the PSM. These results suggest that the PSM contains a specific factor(s) necessary for the destabilization of T-box factors.

## Establishing the PSM-fated cell line using mouse ES cells

To reproduce the in vivo conditions for Tbx6 destabilization, we established a differentiation system for PSM from mouse ES cells. To monitor whether the cells differentiated to PSM, we introduced the *Tbx6-venus* fusion cDNA into the endogenous *Tbx6* locus (*Figure 1E*) via CRISPR/Cas9-mediated homologous recombination (*Feng et al., 2013*). We obtained a recombinant ES cell line in which *Tbx6-venus* was homozygously knocked-in (KI), and used this ES cell line for further experiments.

We modified the conventional ES cell differentiation protocol (*Chal et al., 2015*; *Zhao et al., 2014*) to achieve efficient PSM differentiation. The dissociated feeder-free *Tbx6*$^{Tbx6\text{-}venus/Tbx6\text{-}venus}$ ES cells were cultured in differentiation-medium for 2 days to let the ES cells differentiate to the epiblast state. Then, PSM-fated differentiation was induced by treating the ES cells with 3 μM CHIR99021 (GSK inhibitor) for 2, 3, or 4 days (*Figure 1F*). Compared with ES cells treated with DMSO that exhibited almost no venus-positive cells, ES cells cultured with CHIR99021 exhibited Tbx6-venus expression from day 2 to day 4, as indicated by immunostaining with anti-GFP antibody (*Figure 1G–R*). On day 2, only weak Tbx6-venus signals were observed (*Figure 1N*). The expression of Tbx6-venus peaked at day 3 (*Figure 1P*) and was maintained until day 4 (*Figure 1R*).

## Ripply2-mediated Tbx6 degradation in PSM-fated mouse ES cells

Although we succeeded in inducing Tbx6-positive PSM-fated cells from ES cells, little endogenous Ripply2 expression was detected. We reasoned this to be because Ripply2 expression is induced transiently in the anterior PSM in vivo, which may be a very small population of PSM-fated ES cells. To test whether Ripply2-mediated Tbx6-degradation occurs in the PSM-fated ES cells, we used the Tet-On inducible gene expression system to induce Ripply2 expression upon Doxycycline (Dox) administration. To monitor the expression of both Ripply2 and Tbx6-venus, we designed TRE-*mcherry-T2A-FLAG-Ripply2* in order to detect Ripply2 expression through mcherry expression by fluorescence and by anti-FLAG antibody. The construct was transfected into the *Tbx6*$^{Tbx6\text{-}venus/Tbx6\text{-}venus}$ ES cells (*Figure 2A*). The established *Tbx6*$^{Tbx6\text{-}venus/Tbx6\text{-}venus}$;*mcherry-T2A-FLAG-Ripply2* ES cells were cultured and PSM differentiation was induced by treating with CHIR99021 for 3 days, followed by the addition of 1 μg/mL Dox to induce FLAG-Ripply2 (*Figure 2B*). We confirmed the

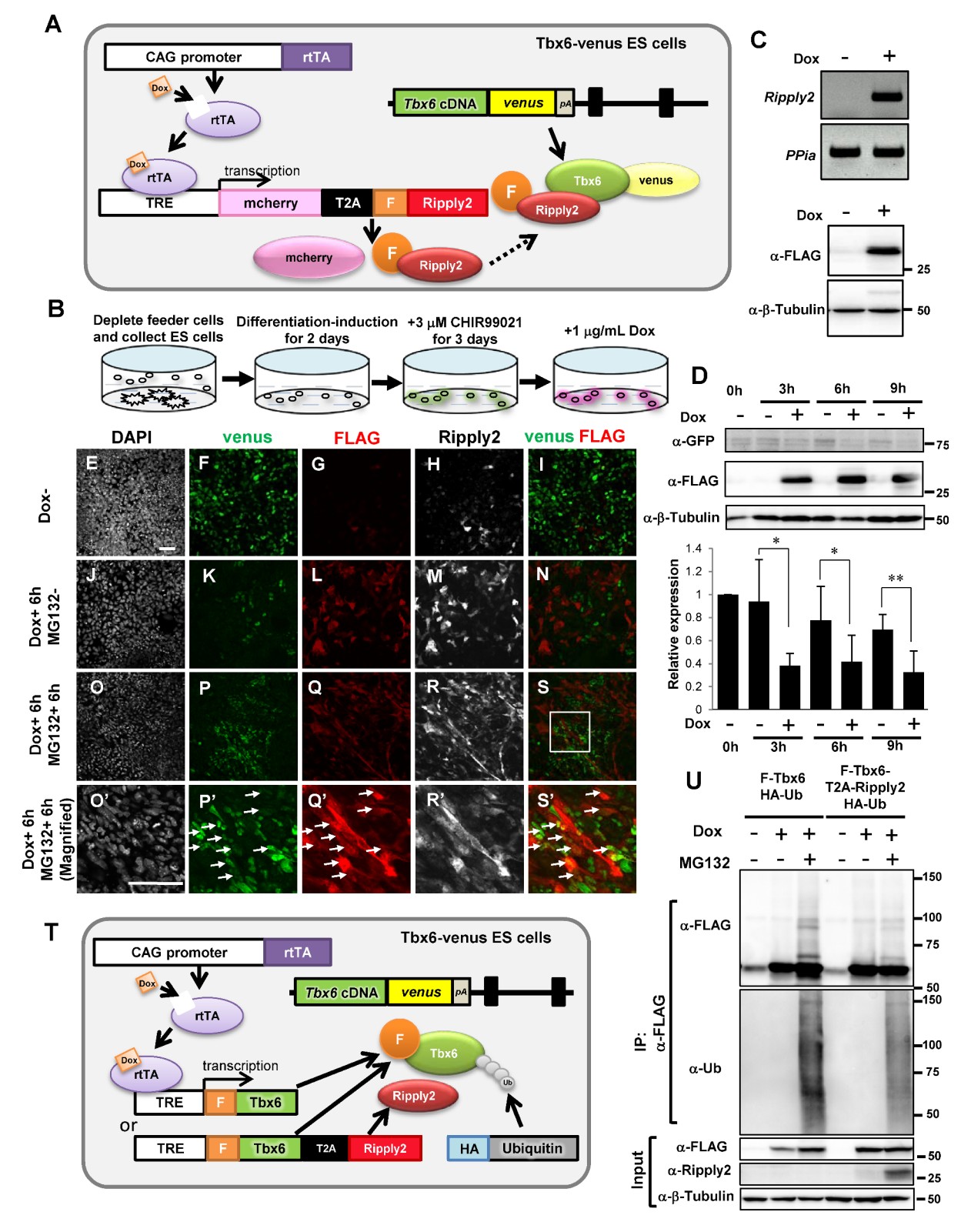

**Figure 2.** Ripply2 repressed Tbx6 protein expression in PSM-fated ES cells. (**A**) Schematic representation of Tet-inducible Ripply2 in Tbx6-venus ES cells. (**B**) Schematic diagram for ES culture. $Tbx6^{Tbx6-venus/Tbx6-venus}$;mcherry-T2A-FLAG-Ripply2 ES cells were cultured under feeder-free conditions for 2 days, followed by culture with 3 μM CHIR99021 for 3 days, and were then treated with 1 μg/ml Dox with or without 10 μM MG132. (**C**) The expression of FLAG-Ripply2 mRNA (up) and protein (down) in $Tbx6^{Tbx6-venus/Tbx6-venus}$ ES cells with Dox treatment for 12 hr without PSM-induction. (**D**) Western blot

*Figure 2 continued on next page*

Figure 2 continued

analyses to monitor Tbx6-venus expression with or without FLAG-Ripply2 expression in PSM-fated ES cells (N = 5). The histogram shows the quantitation of GFP signal normalized by β-Tubulin signal. Asterisks indicate p<0.05, Double asterisks indicate p<0.01; paired t-test. (E–S) Immunofluorescence of PSM-fated ES cells for the detection of Tbx6-venus and FLAG-Ripply2 after 6 hr incubation without Dox (E–I), with Dox (J–N), or with Dox and MG132 (O–S, O'–S'). (O'–S') are the magnified rectangle regions in (O–S). Arrows indicate GFP-FLAG double-positive cells. (N = 3) Scale bar: 50 μm. (T) Schematic representation of Tet-inducible *FLAG-Tbx6;HA-Ubiquitin* or *FLAG-Tbx6-T2A-Ripply2;HA-Ubiquitin* in $Tbx6^{Tbx6-venus/Tbx6-venus}$ ES cells. (U) IP-western analyses of FLAG-Tbx6 immunoprecipitated with anti-FLAG beads from PSM-fated ES cells that expressed FLAG-tagged Tbx6 or FLAG-tagged Tbx6-T2A-Ripply2 with HA-tagged ubiquitin in the absence or presence of MG132 (N = 3).

DOI: https://doi.org/10.7554/eLife.33068.005

The following source data and figure supplements are available for figure 2:

**Source data 1.** Quantification of Tbx6-venus protein in the absence or presence of FLAG-Ripply2 in PSM-fated ES cells.
DOI: https://doi.org/10.7554/eLife.33068.010

**Figure supplement 1.** Decreased Tbx6 protein induced by Ripply2 in the PSM-fated ES cell system.
DOI: https://doi.org/10.7554/eLife.33068.006

**Figure supplement 2.** Proteasome inhibitor interrupted Ripply2-dependant Tbx6 protein degradation in the PSM-fated ES cell system.
DOI: https://doi.org/10.7554/eLife.33068.007

**Figure supplement 3.** Tbx6 is ubiquitinated in the PSM fated cells.
DOI: https://doi.org/10.7554/eLife.33068.008

**Figure supplement 4.** Smurf1/2 are not involved in the Tbx6 degradation mechanism.
DOI: https://doi.org/10.7554/eLife.33068.009

strong induction of both mRNA and FLAG-Ripply2 protein at 12 hr after Dox addition in undifferentiated ES cells (*Figure 2C*). In PSM-fated differentiating ES cells, we found that the expression level of FLAG-Ripply2 reached the peak at 6 hr after induction by Dox (*Figure 2D*), along with a gradual decrease in Tbx6-venus expression from 3 to 12 hr after Dox treatment (*Figure 2D and E–N*, *Figure 2—figure supplement 1A–L*). The disappearance of Tbx6-venus signal appeared to occur quickly in cells with strong Ripply2 signals and slower in areas where the Ripply2 expression level was low (*Figure 2—figure supplement 1A–L*). This Ripply2-dependent reduction in Tbx6 protein was suppressed by the inclusion of 10 μM MG132 (a proteasome inhibitor) (*Figure 2O–S*, *Figure 2—figure supplement 2*), and some Ripply2-Tbx6 double-positive cells were observed under the Dox$^+$-MG132$^+$ conditions (arrows in *Figure 2S'*). These data support the idea that Tbx6 destabilization occurred through the proteasome-mediated degradation pathway.

Next, to investigate whether Tbx6 is ubiquitinated in a Ripply2-dependent manner, we established Tet-inducible ES cell lines that expressed FLAG-Tbx6 or FLAG-Tbx6-T2A-Ripply2 with HA-tagged ubiquitin (*Figure 2T*). We prepared FLAG-Tbx6 via immunoprecipitation using an anti-FLAG antibody before and after PSM induction (*Figure 2U*, *Figure 2—figure supplement 3*). We found several bands with higher molecular weight than FLAG-Tbx6 in the presence of MG132, and the intensity became lower in the presence of Ripply2 (*Figure 2U*), suggesting that these bands were ubiquitinated Tbx6. Ubiquitinated proteins, likely including Tbx6, were observed only in the presence of MG132. Although ubiquitinated Tbx6 was detected even in undifferentiated ES cells, the level of ubiquitination was found to be increased when ES cells were induced to PSM (*Figure 2—figure supplement 3*). These results indicate that Tbx6 can be ubiquitinated even in the absence of Ripply2 in PSM-fated cells and Ripply2 is mainly involved in the degradation step. Regarding the E3 ubiquitin ligase, Smurf1 was reported to be involved in the Smad6-mediated Tbx6 degradation pathway (*Chen et al., 2009*). Therefore, we tested the possible involvement of Smurf, by analyzing Tbx6 expression in *Smurf1/2*-dKO embryos (*Narimatsu et al., 2009*). However, we did not observe any anterior expansion of Tbx6 expression (*Figure 2—figure supplement 4*).

To clarify the necessity of Ripply2-Tbx6 interaction for Tbx6 degradation, we introduced a Tet-inducible *Ripply2$_{ΔFPIQ}$* construct lacking the conserved T-box binding tetrapeptide-FPIQ into $Tbx6^{Tbx6-venus/Tbx6-venus}$ ES cells (*Figure 3A*). As expected from our binding analysis demonstrating that the Tbx6-Ripply2$_{ΔFPIQ}$ interaction was disrupted in HEK293T cultured cells (*Figure 1D*), Tbx6 degradation did not occur in the $Tbx6^{Tbx6-venus/Tbx6-venus}$;*mcherry-T2A-FLAG-Ripply2$_{ΔFPIQ}$* ES cells (*Figure 3B*). We also confirmed no interaction of Tbx6 and Ripply2$_{ΔFPIQ}$ in PSM-fated ES cells (*Figure 3C*). The Ripply2 mutant lacking another conserved motif, WRPW, which is required for interaction with the Groucho transcriptional co-repressors TLE1 and TLE2 (*Figure 3—figure supplement*

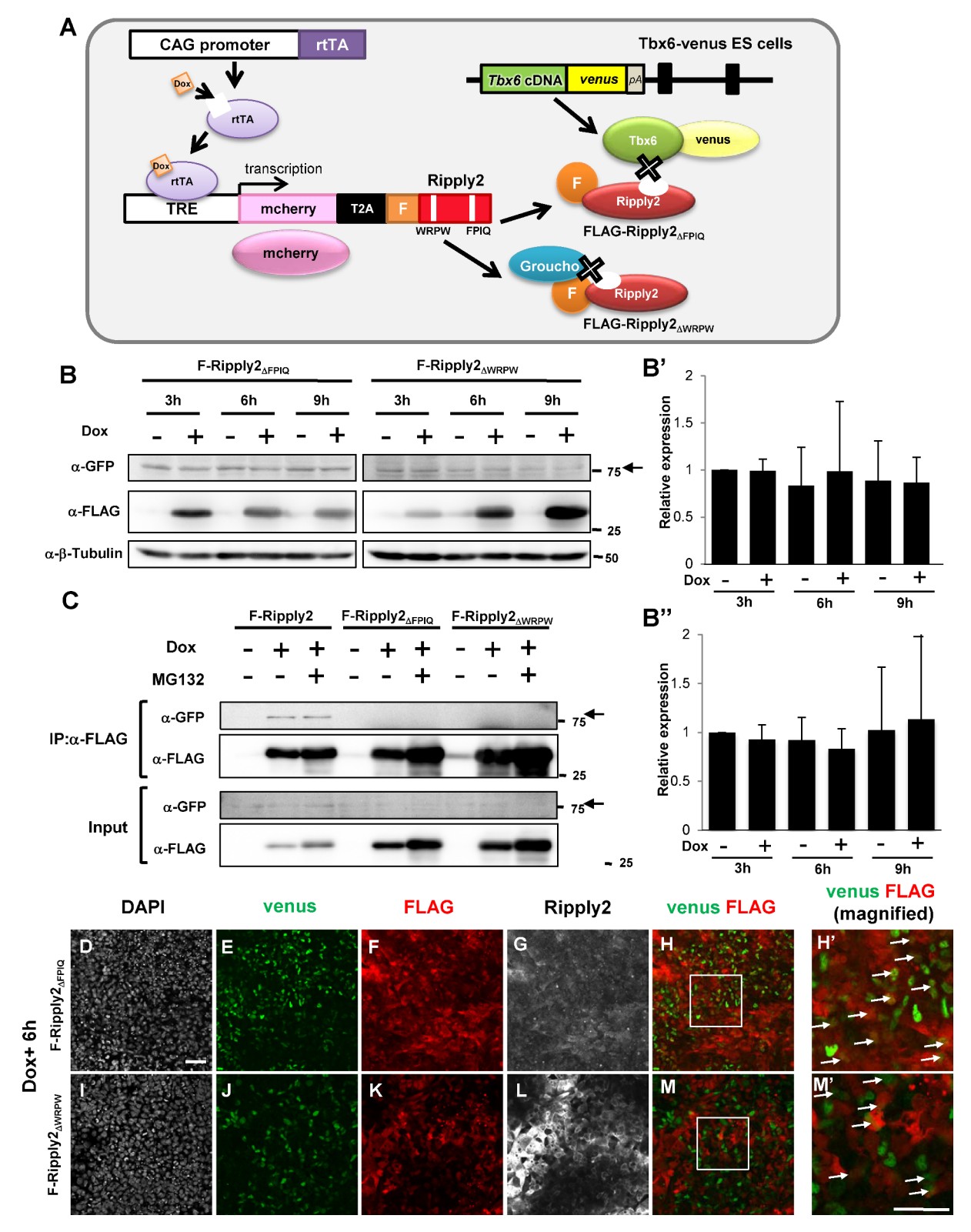

**Figure 3.** Ripply2-mutants lacking interaction with Tbx6 were defective for Tbx6 protein degradation. (**A**) Schematic presentation of Tet-inducible *Ripply2$_{\Delta FPIQ}$* or *Ripply2$_{\Delta WRPW}$* in *Tbx6$^{Tbx6\text{-}venus/Tbx6\text{-}venus}$* ES cells. (**B**) Western blotting of Tbx6-venus in the absence or presence of FLAG-Ripply2$_{\Delta FPIQ}$ or FLAG-Ripply2$_{\Delta WRPW}$ in PSM-fated ES cells. Arrow: Tbx6-venus. The histogram shows the quantitation of GFP signal in Ripply2$_{\Delta FPIQ}$ induced cells (B':
N = 5) and in FLAG-Ripply2$_{\Delta WRPW}$ induced cells (B'':N = 6), normalized by β-Tubulin signal. (**C**) Co-immunoprecipitation of Tbx6-venus using the anti-
*Figure 3 continued on next page*

*Figure 3 continued*

FLAG antibody for FLAG-Ripply2 and the mutants. Tbx6-venus signal was detected using an anti-GFP antibody. Arrows: Tbx6-venus. (N = 5). (**D–M**) Immunofluorescence of PSM-fated ES cells for the detection of Tbx6-venus and FLAG-Ripply2$_{\Delta FPIQ}$ or FLAG-Ripply2$_{\Delta WRPW}$ after 6 hr incubation with Dox. (**H'** and **M'**) are the magnified rectangle regions in (**H** and **M**). Arrows indicate GFP-FLAG double-positive cells. (N = 3) Scale bar: 50 μm.

DOI: https://doi.org/10.7554/eLife.33068.011

The following source data and figure supplement are available for figure 3:

**Source data 1.** Quantification of Tbx6-venus protein in the absence or presence of FLAG-Ripply2DFPIQ or FLAG-Ripply2DWRPW in PSM-fated ES cells.

DOI: https://doi.org/10.7554/eLife.33068.013

**Figure supplement 1.** Ripply2 is able to interact with TLE1/2.

DOI: https://doi.org/10.7554/eLife.33068.012

1), also failed to interact with Tbx6 in cultured cells (*Figure 1D*) and ES cells (*Figure 3C*). As such thereby Tbx6 degradation was also inhibited in PSM-fated ES cells (*Figure 3B*). The inability of both Ripply2$_{\Delta FPIQ}$ and Ripply2$_{\Delta WRPW}$ to degrade Tbx6 was confirmed by immunostaining the PSM-fated ES cells (*Figure 3D–M*). We often observed Tbx6-venus positive signals in Ripply2-expressing cells (*Figure 3H–H', M–M'*). These observations support the idea that degradation of Tbx6 protein depends on the accessibility of Ripply2.

## Ripply2 recruits proteasome complex in the Tbx6 degradation system

To understand the molecular pathway leading to the degradation of Tbx6 via Ripply2, we searched for proteins interacting with Ripply2 via Mass spectrometry analysis. We conducted immunoprecipitation using anti-FLAG antibody conjugated beads to isolate the Ripply2-Tbx6 complex that was formed in the Tbx6 expression/degradation ES cells system (*Figure 4A*). Expression of FLAG-Ripply2 and Tbx6-venus was observed in the input of $Tbx6^{Tbx6-venus/Tbx6-venus}$;*mcherry-T2A-FLAG-Ripply2* ES cell lysate (*Figure 4B* left). After incubating with anti-FLAG beads, both FLAG-Ripply2 and Tbx6-venus protein in the lysate of FLAG-Ripply2 expressing ES cells was precipitated and the corresponding bands disappeared from the supernatant (*Figure 4B* middle, right). Then, we collected the protein complex by eluting the beads with 3xFLAG peptide, and the eluate was used for SDS-PAGE followed by silver staining (*Figure 4C*). Other than the band for FLAG-Ripply2 (*Figure 4C* arrowhead), we observed multiple bands that only existed in the FLAG-Ripply2 expressing panel but not in the control panel, indicating that some specific Ripply2-binding proteins were co-precipitated.

According to mass spectrometry, Ripply2 recruited all subunit components of the 26S proteasome (*Figure 4D*; *Supplementary file 1*), which is known as the main pathway for the degradation of nuclear proteins (*von Mikecz, 2006*). The formation of the Ripply2-proteasome complex was further confirmed by FLAG-Ripply2 immunoprecipitation, followed by western blotting using two kinds of antibodies for proteasome 20S α1, 2, 3, 5, 6 and 7 subunit or 20S core subunit (α5/α7, β1, β5, β5i,β7). The immunoprecipitation data showed that Ripply2 assembled the endogenous proteasome 20S subunit regardless of differentiation induction (*Figure 4E*). Although the antibody for the 20S core subunit was not sensitive enough to detect the endogenous subunits, it successfully detected the subunits accumulated with Ripply2 (*Figure 4E*). In the mass spectrometry result, we also found factors involved in ATP energy metabolism, which may indicate energy exhaustion by the proteasome complex during protein degradation. These results further support our idea that the function of Ripply2 for specific protein degradation is mediated through the proteasome pathway.

## T-box dependent suppression of Tbx6 protein in vivo

In order to further our understanding of the mechanism of Tbx6 degradation, we aimed to identify the amino acid sequences of Tbx6 required for degradation in vivo. To achieve this, we first used BAC (bacterial artificial chromosome)-based transgenic F0 analyses. We introduced the venus-tag into the translational initiation site of *Tbx6* and created BAC-venus, which expressed only venus protein under the control of the Tbx6 promoter (*Figure 5A,B*). We dissected embryos at E10.5, and confirmed that the expression of venus protein was stable and expanded to the anterior region beyond the segmentation point indicated by the arrow (N = 3. *Figure 5B*). We also introduced the venus-tag just before the Tbx6 termination codon to produce the Tbx6-venus fusion protein (*Figure 5A,C*). As expected, the transgene faithfully reproduced the endogenous expression pattern

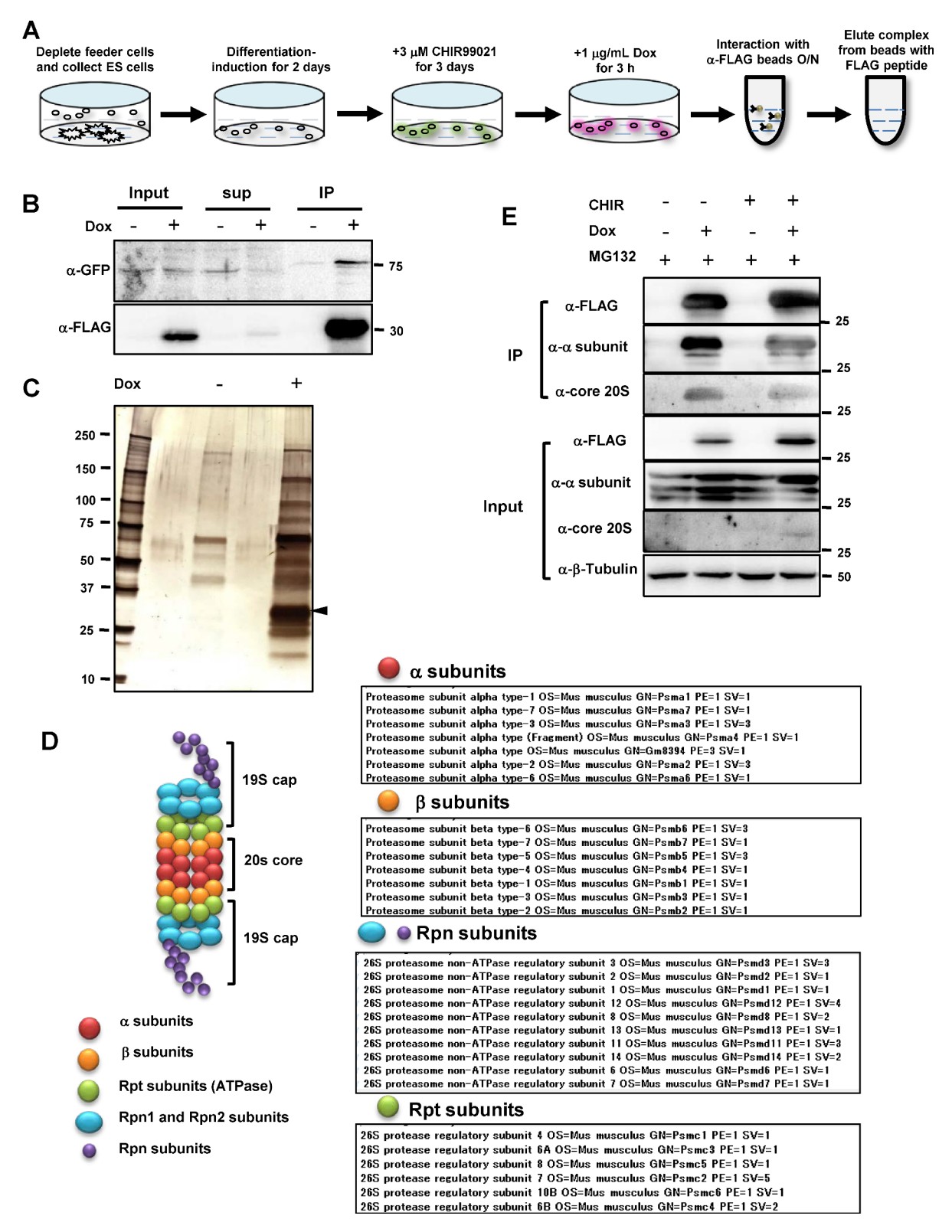

**Figure 4.** MASS-spec analysis of Tbx6-Ripply2 degradation complex using immunoprecipitation of FLAG-Ripply2. (**A**) Schematic diagram of PSM-fated ES culture used for immunoprecipitation. PSM-fated ES cells containing *Tbx6^Tbx6-venus/Tbx6-venus^;mcherry-T2A-FLAG-Ripply2* were incubated with Dox for 3 hr and the cell lysate was subjected to immunoprecipitation. (**B**) Western blotting showing Tbx6-venus and FLAG-Ripply2 in input (left), supernatant after reaction with beads (middle), and after elution with 3xFLAG peptide (right) with anti-GFP and anti-FLAG antibodies. (N = 4) (**C**) Silver staining of
*Figure 4 continued on next page*

*Figure 4 continued*

eluates from beads reacted with control and Tet-induced ES cells lysates. Arrowhead: band for FLAG-Ripply2 protein. (**D**) Proteasome subunit components specifically detected as Ripply2 interacting proteins. (**E**) ES cells cultured under several conditions according to the protocol shown in (**A**) were subjected to immunoprecipitation using anti-FLAG antibody, and interacting proteins were detected with anti-FLAG (Ripply2), anti-proteasome 20 S-1, 2, 3, 5, 6 and 7 subunits, and anti-proteasome 20S core subunit. (N = 2).

DOI: https://doi.org/10.7554/eLife.33068.014

of Tbx6, in which the expression domain has a clear anterior border just before the segmental border in the anterior PSM (N = 3, *Figure 5C*), indicating that the Tbx6-fusion protein was degraded in the anterior PSM. We then inserted the venus-tag after and before the T-box domain to construct fusion proteins lacking only the C-terminal region (BAC-Tbx6$_{\Delta C}$-venus: *Figure 5A,D*) or together with the T-box domain (BAC-Tbx6$_{\Delta TC}$-venus: *Figure 5A,E*), respectively. Tbx6$_{\Delta C}$-venus protein exhibited an expression pattern with a clear anterior border (N = 4, *Figure 5D*); however, the Tbx6$_{\Delta TC}$-venus protein expanded to the anterior region (N = 2, *Figure 5E*). These results indicated that the T-box domain is necessary for degradation of Tbx6 protein. Next, to ask whether the T-box domain itself is sufficient for destabilization, we created BAC-T-box-venus, which expresses only the T-box domain fused with venus (*Figure 5A,F*). Interestingly, the expression of T-box-venus protein had a clear anterior border (N = 5, *Figure 5F*), demonstrating that the T-box region of Tbx6 is necessary and sufficient for degradation. Following immunoprecipitation, only the T-box but not other domains of Tbx6 led to the precipitation of Ripply2 (*Figure 1C*), suggesting that the T-box-dependent Tbx6 degradation is achieved through the interaction between T-box and Ripply2. Using the same method, we further found that the degradation occurred when the T-box was partially deleted until the 152$^{th}$ amino acid (aa) (*Figure 5A,H*, N = 4), but not when it was deleted up to the 124$^{th}$ aa (*Figure 5A,G*, N = 2), suggesting that the 124~152 aa region may be essential for Tbx6 degradation in vivo.

To further examine the region necessary for Tbx6 degradation in vivo, we made chimeras that express Tbx6-venus fusion protein controlled by the endogenous *Tbx6* promoter using the CRISPR/Cas9 system (*Figure 6A*). As Tbx6$_{\Delta 124-152aa}$-venus has no transcriptional activity (*Figure 6—figure supplement 1*), we selected ES clones in which *Tbx6-venus* was heterozygously knocked-in and an endogenous *Tbx6* allele was intact for chimera production (*Tbx6$^{Tbx6(\Delta 124-152aa)-venus/+}$*). We also used ES cells containing an intact *Tbx6* gene fused with *venus* (*Figure 1E*: *Tbx6$^{Tbx6-venus/Tbx6-venus}$*) as a positive control. We analyzed chimera embryos on embryonic day 10.5 (E10.5). The negative control, wild-type embryos, had only background signal for venus detection (N = 16, *Figure 6B*), whereas the positive control exhibited fluorescent signal with a sharp anterior limit as well as endogenous Tbx6 protein expression (N = 18, *Figure 6C*), suggesting Tbx6-venus as suitable to represent the expression of Tbx6 protein. Intriguingly, the Tbx6-venus signal in the chimera with *Tbx6$^{Tbx6(\Delta 124-152aa)-venus/+}$* ES cells became stabilized and expanded anteriorly (N = 20, *Figure 6D*). We then conducted whole-mount triple immunostaining to compare the expression pattern of Tbx6-venus protein with endogenous Tbx6 protein by comparing the images using anti-GFP and anti-Tbx6 antibodies. It should be noted that the anti-Tbx6 antibody produced against the Tbx6 C-terminal peptide did not recognize the Tbx6-venus fusion protein. We confirmed that Tbx6-venus in chimera embryo with *Tbx6$^{Tbx6(\Delta 124-152aa)-venus/+}$* ES cells had an expanded pattern (*Figure 6E*), whereas endogenous Tbx6 showed a clear anterior limit that was at the same position as the posterior expression domain of Ripply2 (*Figure 6E–H*), indicating that endogenous Tbx6 but not mutant Tbx6 protein was degraded by Ripply2 in the chimera embryos. These observations demonstrated that the T-box, especially the aa125~152 domain, plays a central role in the degradation of Tbx6 protein in vivo.

To examine whether the stabilized version of Tbx6 was due to the loss of interaction with Ripply2, we generated several deletion forms of the Tbx6 protein similar with those used in the in vivo experiments and performed co-immunoprecipitation assay with Ripply2. As expected, when the C-terminal from the 124$^{th}$ amino acid was deleted from Tbx6 (Tbx6$_{1~124aa}$), the mutant protein failed to bind Ripply2 (*Figure 6I–J*), but Tbx6$_{1~152aa}$ succeeded in interacting with Ripply2 (*Figure 6I–J*), suggesting that amino acids 125 ~ 152 contain the Ripply2-binding-motif in the T-box. However, we surprisingly found that the deletion of 125 ~ 152 aa (Tbx6$_{\Delta 124~152}$) did not affect the ability to form a complex with Ripply2 either in HEK293T immunoprecipitation analysis or in GST-pull-down assay

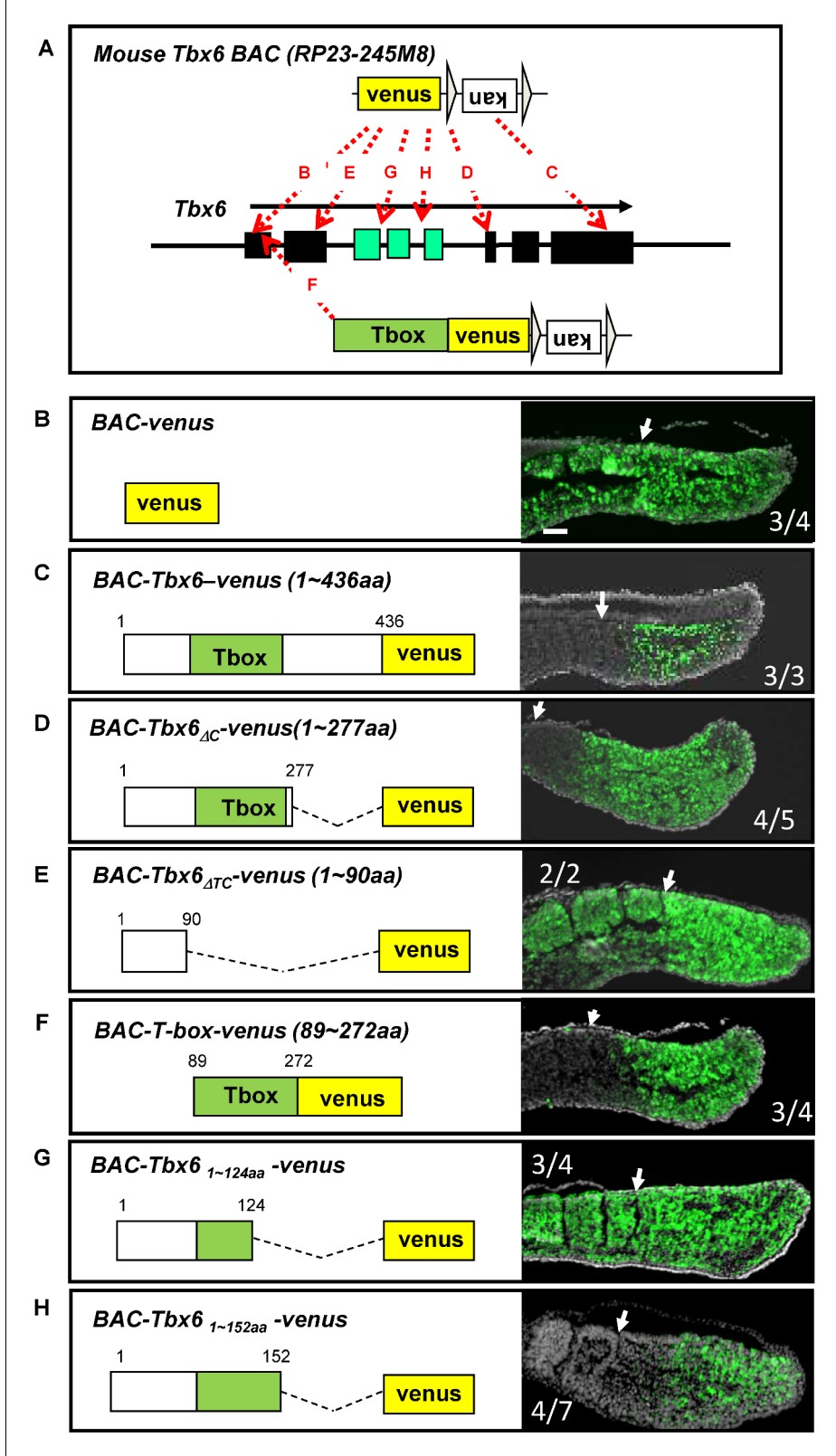

**Figure 5.** The T-box is essential and sufficient for Tbx6 destabilization in vivo. (**A**) Strategies to generate BAC constructs with venus-tag at different positions in the Tbx6 protein. The venus-tag was introduced in frame with either the translational initiation site (B, BAC-venus), translational termination site (C, BAC-Tbx6-venus), after T-box (D, BAC-Tbx6$_{\Delta C}$-venus) or before the T-box (E, BAC-Tbx6$_{\Delta TC}$-venus), after amino acid 124 within the T-box (G,

*Figure 5 continued on next page*

*Figure 5 continued*

BAC-Tbx6$_{1-124aa}$ -venus, amino acid 152 (H, BAC-Tbx6$_{1-152aa}$ -venus). A similar method was also used to generate a construct containing only the T-box with venus (F, BAC-T-box-venus). Black and green boxes indicate exons in the *Tbx6* locus. Green ones correspond to the T-box region. (B–H) Sections of E10.5 transgenic mouse embryos harboring each BAC construct were stained with anti-GFP antibody. Construct names and amino acid sequences included in the Tbx6-venus fusion-proteins are indicated in each panel. Green; venus signal. Gray; DAPI staining. Newly formed somite borders are indicated by white arrows. Numbers of GFP-positive embryos among transgenic embryos recovered are shown within each box. Scale bar: 100 μm.

DOI: https://doi.org/10.7554/eLife.33068.015

(*Figure 6I–K*), suggesting that although 125 ~ 152 aa has Ripply2-binding ability it is not the only Ripply2-binding motif in the T-box. Therefore, the stabilization of Tbx6$_{\Delta124-152aa}$-venus in the chimera embryo was not due to disruption of Ripply2 association but perhaps to a special motif for degradation in the region of 125 ~ 152 aa.

## Discussion

### Recapitulation of Tbx6-degradation using the PSM-fated ES differentiation system

The T-box family of transcription factors play crucial roles throughout embryonic development and their expression patterns are strictly controlled via tissue specific context-dependent mechanisms. The activation/inactivation switch for Tbx6 via the expression of Ripply2 in the anterior PSM is a key issue for the determination of the segmental border during somitogenesis. However, it is almost impossible to assess the biochemical nature of the Ripply2-dependent Tbx6 degradation mechanism using in vivo samples because the segmental border is an ephemeral structure that contains only a small cell population. To solve this problem, we established a PSM-fated ES cell system, which can provide a similar environment to in vivo, and successfully reproduced the Tbx6 expression/degradation process using a very simple method. We introduced a venus tag just before the stop codon in the endogenous *Tbx6* gene using a Cas9/CRISPR-mediated method to make the Tbx6-venus fusion protein, which enabled us to identify specific cells in the PSM, and importantly, allowed us to monitor Tbx6 protein stability during the differentiation process. Using this monitoring system, we observed the quick disappearance of Tbx6-venus signal upon Ripply2 expression. The reduction of Tbx6-venus occurred in a short time; the effective time was less than 2 hr, which is comparable with the time observed in in vivo somitogenesis. This system also allowed us to prepare Ripply2-interacting proteins required for Tbx6 degradation.

### Ripply2 mainly functions as a factor involved in protein degradation in mice

Activation and termination of transcription factor activity in specific tissues at an appropriate time is important during embryonic development. The termination of Mesp expression is a pivotal pathway in segmental border determination in mice, but the regulation cascade of Tbx6, Mesp and Ripply likely diverged during vertebrate evolution. Zebrafish tbx6 directly induces the expression of ripply without mediating mesp function, after which ripply1/2 in turn suppresses the expression of both tbx6 and mesp (*Wanglar et al., 2014*; *Windner et al., 2015*; *Yabe et al., 2016*). On the other hand, mouse Ripply2 requires Mesp2 for transcriptional activation in addition to Tbx6 (*Dunty et al., 2008*; *Morimoto et al., 2007*), and suppresses Mesp2 function through destabilization of Tbx6. Repression of transcriptional activity of T-box family transcriptional factors is a well-known function of Ripply family factors in zebrafish and *Xenopus* (*Kawamura et al., 2005*; *Kawamura et al., 2008*; *Kondow et al., 2007*). However, there is no evidence of a similar repression mechanism in mice, and we suggested the negative regulation in mouse to be mediated by the destabilization of T-box proteins (*Oginuma et al., 2008*; *Zhao et al., 2015*), arguing that Ripply factors are not completely functionally conserved within vertebrates. This study provided clear evidence that Ripply2 participates in Tbx6 protein degradation through the proteasome pathway. Proteomic screening of Ripply2-binding proteins from PSM-fated mouse ES cells identified all members of 26S proteasome subunits, and

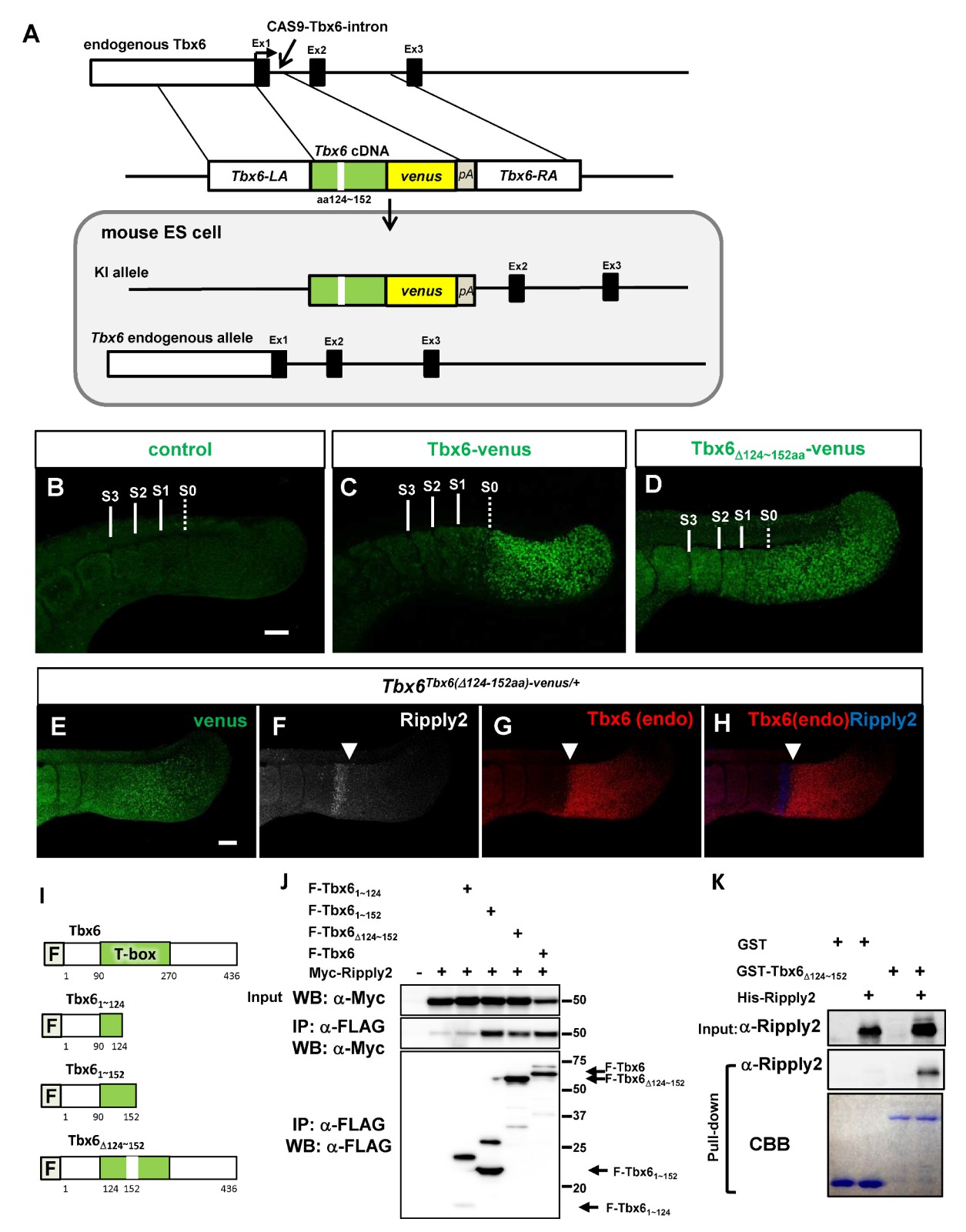

**Figure 6.** Identification of T-box subdomain required for the degradation of Tbx6 *via* Ripply2. (**A**) The strategy for generating $Tbx6^{Tbx6(\Delta124\text{-}152aa)\text{-}venus/+}$ + cells. (**B–D**) Whole-mount immunofluorescent image for venus signal of WT (**B**) and chimera embryos produced with $Tbx6^{Tbx6\text{-}venus/Tbx6\text{-}venus}$ (**C**), and $Tbx6^{Tbx6(\Delta124\text{-}152aa)\text{-}venus/+}$ (**D**) ES cells at E10.5. (**E–H**) Whole-mount triple-immunostaining for the chimera embryo produced with $Tbx6^{Tbx6(\Delta124\text{-}152aa)\text{-}venus/+}$ cells showing anti-GFP (**E**), anti-Ripply2 (**F**), and anti-Tbx6 (**G**), and the merged (**H**, Tbx6 and Ripply2) signals. Expanded GFP signal but not

*Figure 6 continued on next page*

*Figure 6 continued*

endogenous Tbx6 signal was observed in the chimera embryo with Tbx6^{Tbx6(Δ124-152aa)-venus/+} cells. Arrow heads: anterior limit of the endogenous Tbx6/ posterior limit of Ripply2. Scale bar: 100 μm. (I) Schematic presentation of Tbx6 constructs used for binding assays shown in (J) and (K). (J) IP-western analyses. Each FLAG-tagged Tbx6 construct was co-transfected with Myc-Ripply2 into HEK293T cells. The lysates were subjected to IP with anti-FLAG antibody, followed by western blot analyses. (N = 5) (K) GST-pull down assay. GST or GST-Tbx6_{Δ124-152aa} was incubated with purified His-Ripply2 and pulled down with Glutathione Sepharose, and then subjected to western blotting. His-Ripply2 was detected by anti-Ripply2 antibody. CBB: Coomassie Brilliant Blue staining. (N = 2).

DOI: https://doi.org/10.7554/eLife.33068.016

The following source data and figure supplements are available for figure 6:

**Figure supplement 1.** Tbx6_{Δ124−152aa} has no transcriptional activity.

DOI: https://doi.org/10.7554/eLife.33068.017

**Figure supplement 1—source data 1.** Quantification of Tbx6-venus protein in the absence or presence of FLAG-Ripply2 in PSM-fated ES cells.

DOI: https://doi.org/10.7554/eLife.33068.018

Ripply2-Tbx6 was able to form a complex through direct interaction, demonstrating that Ripply2 physically associates with the proteasome anβd targets Tbx6 protein to the proteasome for permanent inactivation by degradation. Ripply2-mediated Tbx6 degradation was interrupted by the proteasome inhibitor MG132. This result is in agreement with the in vivo discovery that the Tbx6 expression domain was elongated by one-somite-length with MG132 treatment for 2 hr (*Oginuma et al., 2008*).

Although TLE1/2 are well-known Groucho family co-repressors and their expression was observed in mouse PSM tissue, mouse Tbx6 transcriptional activity was never affected when TLE1 and Ripply2 were co-transfected in an Mesp2 luciferase reporter system (*Figure 1—figure supplement 1*). This raises the question of how TLE1/2 functions in mouse PSM tissue. We found that the Groucho-binding motif of Ripply2 is also required for binding to Tbx6, indicating that TLE1/2 may participate in the degradation of Tbx6 via protein-protein interaction. Mass spectrometry analysis also revealed TLE in the Ripply2-Tbx6 complex, possibly reflecting the involvement of TLE in Tbx6 degradation in the PSM, but its molecular function in protein degradation remains unclear.

It is of note that Ripply2 can recruit the 26S proteasome subunit in ES cells without PSM-induction (*Figure 3E*), even though the Ripply2-mediated T-box factor degradation was found to be a highly context-dependent event. Tbx6 degradation never occurred in cultured cells, and even in vivo, the Ripply2-mediated induction of T-box factor degradation occurred only in PSM tissue (*Zhao et al., 2015*) but not in other mesodermal tissue such as the heart (*Figure 1—figure supplement 2*). This suggests that other than proteasomes, there are other factor(s), possibly cell-specific E3 ligases, that help to down-regulate Tbx6 in PSM tissue. Consistent with this observation, Tbx6 was ubiquitinated only at a low level in undifferentiated ES cells (*Figure 2—figure supplement 3*). Once the ES cells were induced to PSM, the ubiquitination was accelerated even in the absence of Ripply2. This suggests that Ripply2 is not involved in the ubiquitination step and only in the degradation step of Tbx6, indicating that a PSM-specific E3 ligase must be induced and functions in Tbx6 ubiquitination. However, such a candidate protein was not included in our MASS analysis. Additional studies are needed to identify the responsible E3 ligase and/or unknown factor(s) involved in Tbx6 degradation in the PSM.

## T-box is A conserved motif for recognition by Ripply2 during degradation

The T-box, which spans 180–200 amino acid residues, is a relatively large domain for the requirement of well-known DNA binding functions, suggesting that other functions likely depend on the T-box. For example, the β-sheet structures can constitute protein-protein interfaces for binding of other factors. In our study, the T-box domain was confirmed to be necessary and sufficient for both Ripply2-binding and Tbx6 degradation. The T-box family factors are ancient in origin and present in all metazoans, calling into question whether Ripply-dependent destabilization is a general event for other T-box factors in other species. Besides Tbx6, other T-box family transcriptional factors, such as T (Brachyury), have also been found to be degraded by Ripply2 (*Zhao et al., 2015*). Zebrafish tbx6 protein has been reported to be destabilized *via* ripply1/2 expression (*Wanglar et al., 2014*). These studies support the idea that the T-box dependent degradation pathway is conserved among T-box

family factors. The domain of 124 ~ 152 aa in the T-box was found to be necessary for Tbx6 degradation in our chimera mouse experiment. However, Tbx6$_{\Delta124-152}$ did not lose its ability to bind Ripply2, indicating that this region regulates degradation not by Ripply2-Tbx6 interaction but through providing a critical protein degradation point, such as a ubiquitin-binding site. It has been reported that ubiquitin is usually attached to the ε-amino group of lysine through an isopeptide bond (*Hicke et al., 2005*). However, no lysine residue or ubiquitin-binding domain was found in the 124 ~ 152 aa region. This means that this T-box region likely contains an unusual ubiquitylation site or a specific factor may be required for the degradation of T-box factors. There is much more to be learned about the co-factors in the degradation process of T-box factors to fully understand the tissue-specific developmental regulation mechanisms during embryogenesis. Research using PSM-fated ES cells offers a rich source of material that will lead to further understanding of these molecular mechanisms.

## Materials and methods

### The strategy to generate the expression vectors used for biochemical analyses

The wild-type Tbx6 (*Oginuma et al., 2008*) and Ripply2 (*Morimoto et al., 2007*) cDNA were cloned in vectors shown in *Supplementary file 2*. The mutant constructs of Tbx6 and Ripply2 were generated by methods shown in *Supplementary file 3*.

### Cell lines

In this study, we used cultured cell lines HEK293T and Cos7, and several modified ES cell lines originated from TT2 ES cell lines (*Yagi et al., 1993*). These cell lines were free of mycoplasma based on MycoAlert Mycoplasma Detection Assay (Lonza, TX, USA).

### Immunoprecipitation and western blot

HEK293T cells, obtained from ATCC (Manassas, VA), were transiently transfected with DNA constructs *via* PEI (Polyethylenimine) (polysciences, Inc, PA, USA), and then incubated in DMEM supplemented with 10% fetal bovine serum for 24 hr before analysis.

For immunoprecipitation, cells were lysed in 50 mM Tris-HCl (pH 7.4), 150 mM NaCl, 1 mM EDTA (pH 8.0), 0.5% Nonidet P-40, 1 mM dithiothreitol (DTT) and cOmplete EDTA-free Protease Inhibitor Cocktail (Sigma-Aldrich, Germany). Immunoprecipitation was performed as described previously (*Lou et al., 2006*). Briefly, the supernatants were incubated with 10 µl of anti-FLAG M2 affinity gel (Sigma,MO, USA) on a rotator at 4°C. After several washes, precipitates were boiled with 3 × Sample buffer, separated by SDS-PAGE, and then subjected to western blotting analysis as described previously (*Zhao et al., 2015*). Western analysis was performed using the primary antibodies; rabbit anti-Myc antibody (1/5000, Sigma), mouse anti-FLAG (1/5000, Sigma), mouse anti-FLAG-HRP (1/3000, Sigma), rabbit anti-Ripply2 (1/1000), rabbit anti-GFP (1/2000, MBL, JAPAN), mouse anti-Proteasome 20S α1, 2, 3, 5, 6 &and 7 subunits (1/5000, Enzo Life Sciences, NY, USA), rabbit anti-Proteasome 20S core subunits (1/2000, Enzo Life Sciences), mouse anti-Mono- and polyubiquitinylated conjugates monoclonal antibody (FK2)–HRP (1/1000, Enzo Life Sciences) and mouse anti-β-tubulin (1/5000, Sigma), followed by incubation with goat anti-rabbit IgG conjugated with HRP (1/5000, cell signaling, MA, USA) and donkey anti-mouse IgG conjugated with HRP (Jackson Immuno Research, PA, USA) as the secondary antibodies. Quantification of western blot analyses were conducted using Analyze Gel function of ImageJ software.

### GST pull-down assay

His-tagged Ripply2 and the mutant proteins were expressed in E. coli, BL21, and purified with TALON Metal Affinity Resin (BD Bioscience, CA, USA). The GST pull-down method was described previously (*Suzuki et al., 2012*). Briefly, GST-tagged Tbx6 and the mutant proteins were expressed in the E. coli BL21 (DE3) strain and the bacterial pellets were sonicated in binding buffer (25 mM HEPES-KOH [pH 7.4], 150 mM NaCl, 0.1% NP-40, 1 mM DTT, 1 mM EDTA, 1 mM PMSF), and then spun at 15,000 rpm at 4°C. The supernatants were mixed with His-Ripply2 and mixed with 30 ml of glutathione-sepharose 4FF (GE Healthcare, Sweden) followed by incubation for 2 hr. After extensive

washing, precipitates were separated by SDS-PAGE and analyzed by western blotting with anti-Ripply2 antibody or by CBB staining.

## Animals

All mice were handled and propagated in accordance with National Institute of Genetics (NIG) guidelines, and all experimental procedures were approved by the Committee for Animal Care and Use in NIG. Staging of animals was done by designating the day on which a copulatory plug was detected as E0.5 and the day of birth as P0.5. All mice used in this study were maintained in MCH background (CLEA Japan). The mouse lines used in this study were previously described: *CAG-floxed lynmRFP fused with FLAG-Ripply2* (*Zhao et al., 2015*), *Mesp1-Cre* (*Saga et al., 1999*), *Smurf1/2-DKO* (*Narimatsu et al., 2009*).

## Targeting strategy to generate $Tbx6^{Tbx6\text{-}venus/Tbx6\text{-}venus}$, $Tbx6^{Tbx6(\Delta124\text{-}152aa)\text{-}venus/+}$ KI ES cells using the Cas9/CRISPR system

To construct the *Tbx6-venus* targeting vector, Tbx6 cDNA without a stop codon was ligated into the N-terminal of the venus (ATG removed)-pCS2$^+$ vector, kindly provided by Atsushi Miyawaki (RIKEN). A 1 kb genomic DNA fragment of the 5' upstream region (LA) and 1 kb genomic DNA fragment from 3' downstream of the Cas9 target site of *Tbx6* (RA) were ligated with *Tbx6-venus* in the pCS2 vector. For generating Tbx6 mutant targeting vectors, full-length *Tbx6* cDNA was replaced with $Tbx6_{\Delta124-152aa}$ in the *Tbx6-venus*-targeting vector.

The bicistronic expression vector expressing sgRNA and hCas9 mRNA (pX330) (*Cong et al., 2013*) was purchased from Addgene (Cambridge, MA) and a puromysin$^r$ cassette was inserted. The Cas9-target sequence designed for the *Tbx6-intron-1* (5'-caccGTGAGCGGTTGGATTGGCTC-3' and 5'- aaacGAGCCAATCCAACCGCTCAC −3') was annealed and ligated in to the px330-puro vector. These vectors were introduced into TT2 ES cells (C57BL/6 (B6)/CBA) by transfection using Lipofectamine 2000 (Invitrogen). After selection using puromycin for 24–36 hr, resistant clones were isolated and their DNA was analyzed by PCR using primers, Tbx6-GL1: (5'−CAGAGAGGGGACCTGGAATCC −3'), and Tbx6-R primer: (5'−CTCGTGGATGGTACATGTTGTACCG−3') for the 5'-terminal; *NeoAL2* (5'−GAAAGAACCAGCTGGGGCTCGAG−3'), and Tbx6-GR1: (5'−GCCCCTTCACTCTCTCCATCC TAG−3') for the 3' terminal.

## Introduction of Tet-on system in ES cells

To establish the Tet-inducible Ripply2, Tbx6, Tbx6-T2A-Ripply2-expression systems, a piggy bac transposon system was used as previously described (*Li et al., 2013*), *pBase*, *CAG*-promoter-driven rtTA and either *pPB-CMV-mcherry-T2A-FLAG-Ripply2*, *pPB-CMV-mcherry-T2A-FLAG-Ripply2$_{\Delta FPIQ}$*, *pPB-CMV-mcherry-T2A-FLAG-Ripply2$_{\Delta WRPW}$*, *pPB-CMV-FLAG-Tbx6 or pPB-CMV-FLAG-Tbx6-T2A-Ripply2* vectors were transfected into the *Tbx6-venus* ES cells using Lipofectamine 2000. The ES cells were selected using neomycin. To detect ubiquitinated Tbx6, *pPB-CMV-HA-Ubiquitin* was additionally introduced into the $Tbx6^{Tbx6\text{-}venus/Tbx6\text{-}venus}$ ES cells containing either *pPB-CMV-FLAG-Tbx6 or pPB-CMV-FLAG-Tbx6-T2A-Ripply2.*

## ES culture and PSM induction methods

ES cells containing homozygous *Tbx6-venus* knock-in alleles with Tet-inducible transgenes were cultured with feeder cells in ES medium (*Yagi et al., 1993*).

For PSM induction, the feeder cells were depleted and ES cells were cultured on gelatin–coated culture dishes for western analysis, or on human fibronectin-coated coverglasses for immunostaining with differentiation medium (2% L-Glutamine (Life Technologies, Brazil), 1xMEM NEAA (Life Technologies, NY, USA), 10 units/ml Penicillin 10 μg/ml Streptomycin (Life Technologies), 55 μM 2-Mercaptoethanol, 8 μg/ml Adenosine (SIGMA-Aldrich, China), 7.3 μg/ml Cytidine (SIGMA-Aldrich) 8.5 μg/ml Guanosine (SIGMA-Aldrich), 2.4 μg/ml Thymidine (SIGMA-Aldrich), 7.3 μg/ml Uridine (SIGMA-Aldrich), and 10% FBS (SIGMA-Aldrich, MI, USA) in DMEM (Life Technologies, UK) for 2 days, followed by culture with differentiation medium containing 3 μM CHIR99021 (Wako, JAPAN) for 3 days. To induce the Tet-inducible expression system, 1 μg/mL Doxycycline (SIGMA-Aldrich, Switzerland) was added into the medium for varying periods.

## Immunostaining

Immunostaining for cultured cells was performed on cover glasses coated with human fibronectin. Cells was fixed by 4%PFA on ice for 10 min, then blocked using 3% skim milk, followed by incubating with primary antibodies; mouse-anti-FLAG(M2) (1/5000, Sigma), guinea pig anti-Ripply2 (1/100) (*Zhao et al., 2015*), and chick anti-GFP (1/400, Aves Labs, Oregon, USA) at 4°C overnight and then incubated with secondary antibodies; Alexa Fluor 488-conjugated anti-rabbit antibody (1/800, Life Technologies, Oregon, USA), Alexa Fluor 594-conjugated anti-mouse antibody (1/800, Life Technologies), Cy5 conjugated anti-Guinea pig antibody. The methods used for whole mount immunostaining and section immunostaining are described in our previous reports (*Zhao et al., 2015*). Antibodies used in this study, anti-Tbx6 and anti-Ripply2, were described previously (*White and Chapman, 2005*; *Zhao et al., 2015*). For immunostaining of heart sections, we used rabbit-anti-TBX5 (H-70)(1/300, Santa Cruz Biotechnology, CA, USA), goat-anti-TBX18 (C-20) (1/300, Santa Cruz Biotechnology) as first antibodies. These sections were observed using FluoView FV1200 laser scanning confocal microscopy (Olympus).

## Mass spectrometry for analyzing Ripply2-interacting proteins

$Tbx6^{Tbx6-venus/Tbx6-venus}$;*mcherry-T2A-FLAG-Ripply2* ES cells were induced to undergo differentiation toward a PSM-fate by culturing with 3 µM CHIR99021, followed by 1 µg/mL Dox for 3 hr to induce the expression of FLAG-Ripply2. The lysate from PSM-fated ES cells was incubated with anti-FLAG beads overnight, and the beads were washed with washing-buffer (50 mM Tris-HCl (pH 7.4), 150 mM NaCl, 1 mM EDTA (pH 8.0), 0.5% Nonidet P-40, 1 mM dithiothreitol (DTT)) for at least eight times, then the proteins interacting with FLAG-Ripply2 were collected by eluting with 150 µg/mL 3xFLAG peptide (SIGMA-Aldrich). The beads were spun down and the supernatant was applied to a gradient acrylamide gel (Bio-Rad, USA). Mass spectrometry was conducted by the Riken Center for Developmental Biology.

## Generation of BAC transgenic mice

BAC DNA modifications were conducted using the λ red recombination method as described previously (*Datsenko and Wanner, 2000*; *Oginuma et al., 2008*). Briefly, we utilized 70 nt primers with 50 nt of homology to the insert point of the *Tbx6* gene and 20 nt of homology to the recombination vectors containing venus or T-box-venus followed by an FRT-flanked kanamycin resistance cassette (*Figure 5A*). The resulting PCR products were introduced into competent DH10 BAC (*PR23-245M8*) host cells containing *PKD46*, which carries the λ recombination genes. The recombinants were characterized using specific PCR. The kanamycin resistance cassette was removed by introduction of a *pCP20* plasmid containing temperature-sensitive FLP.

Transgenic mice were generated by microinjection of each BAC vector into fertilized eggs. Microinjected eggs were then transferred into the oviducts of pseudopregnant foster mothers. The genotypes of the embryos were identified by PCR using yolk sac DNA.

## Chimera analysis

For chimera production, two kinds of ES cells were used. One was control ES cells containing homozygous *Tbx6-venus* in both Tbx6 alleles ($Tbx6^{Tbx6-venus/Tbx6-venus}$) (*Figure 1E*) and the other was mutant ES cells containing *Tbx6 (Δ124 ~ 152 aa)-venus* in a single *Tbx6* locus ($Tbx6^{Tbx6(Δ124-152aa)-venus/+}$) (*Figure 6*). ES cells were aggregated with 8 cell stage embryos and the formed blastocysts were transferred into the uterus of a pseudopregnant foster mother the next day. After 8 days (corresponding to E10.5), embryos were recovered and examined for Tbx6-venus expression. Tbx6-venus positive embryos were fixed and processed for whole mount immunostaining using anti-GFP, anti-Ripply2, and anti-Tbx6 antibodies. The confocal images were obtained using Olympus FV1200.

## Luciferase assay

The luciferase assay was performed as previously described (*Yasuhiko et al., 2006*). The vectors for control, NICD, Tbx6, Tbx6 + NICD, and $Tbx6_{Δ124~152aa}$ + NICD were transfected into Cos7 cells for 24 hr, respectively. The cell lysates were subjected to a luciferase assay using the Dual Luciferase System (Promega, WI, USA). The experiments were performed in triplicate for each assay.

## Acknowledgements
We are grateful to Dr. Hitoshi Niwa for providing us piggy bac vectors and to Dr. Yukuto Yasuhiko for providing reagents and the luciferase assay data for examining the influence of TLE1 on the transcriptional activity of Tbx6. Dr. Atsushi Miyawaki (RIKEN) kindly provided the plasmid pCS2-venus. We also thank Drs. Masahiro Narimatsu and Jeffrey L Warana for providing Smurf1/2 double KO embryos. We are also grateful to Yuko Sakakibara for technical support and Danelle Wright for editing this manuscript.

## Additional information

### Funding

| Funder | Author |
| --- | --- |
| Transdisciplinary Research Integration Center of the Research organization of Information and System | Yumiko Saga |

The funders had no role in study design, data collection and interpretation, or the decision to submit the work for publication.

### Author contributions
Wei Zhao, Conceptualization, Supervision, Methodology, Writing—original draft, Project administration, Writing—review and editing; Masayuki Oginuma, Conceptualization, Validation, Investigation, Visualization, Methodology, Writing—original draft, Writing—review and editing; Rieko Ajima, Resources, Investigation, Visualization, Methodology, Writing—original draft; Makoto Kiso, Validation, Investigation, Visualization, Methodology, Writing—review and editing; Akemi Okubo, Resources, Investigation, Methodology; Yumiko Saga, Resources, Validation, Investigation, Visualization

### Author ORCIDs
Yumiko Saga (iD) http://orcid.org/0000-0001-9198-5164

### Ethics
Animal experimentation: This study was performed in strict accordance with the recommendations in the Guide for the Care and Use of Laboratory Animals of the National Institutes of Genetics (NIG). All of the animals were handled according to approved institutional animal care and use committee protocols of NIG. The protocol was approved by the Committee on the Ethics of Animal Experiments of NIG (Permit Number: 29-8 and 29-9). All surgery was performed under anesthesia, and every effort was made to minimize suffering.

### Decision letter and Author response
Decision letter https://doi.org/10.7554/eLife.33068.024
Author response https://doi.org/10.7554/eLife.33068.025

## Additional files
### Supplementary files
• Supplementary file 1. Ripply2-interecting proteins identified by Mass Spectrometry. Identified proteins, which spectrum counts are higher in Flag IP than control, are listed. Ripply2 is highlighted with yellow, Proteasome subunits are highlighted with green, and Tbx6 is highlighted with pink.
DOI: https://doi.org/10.7554/eLife.33068.019

• Supplementary file 2. Vector information used for each construct. Vector information of cDNA constructs used for immunoprecipitation experiments are indicated. The corresponding Figures obtained by using each construct are also listed.
DOI: https://doi.org/10.7554/eLife.33068.020

• Supplementary file 3. Strategies to generate mutant constructs. Primer information and the cloning strategies are listed for each mutant cDNA constructs.
DOI: https://doi.org/10.7554/eLife.33068.021

• Transparent reporting form
DOI: https://doi.org/10.7554/eLife.33068.022

### Data availability

All data generated or analysed during this study are included in the manuscript and supporting files. Source data files have been provided for Figures 2D, 3B and Figure 6-Figure supplement 1. Key resource table is also provided.

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
