## [Decision Letter]

Thank you for submitting your article "Ripply2 recruits proteasome complex for Tbx6 degradation to define segment border during murine somitogenesis" for consideration by *eLife*. Your article has been reviewed by two peer reviewers, and the evaluation has been overseen by a Reviewing Editor and Marianne Bronner as the Senior Editor. The reviewers have opted to remain anonymous.

The reviewers have discussed the reviews with one another and the Reviewing Editor has drafted this decision to help you prepare a revised submission..

The regular formation of somites, that takes place progressively on the anterior/posterior axis of the embryo, is a fascinating process. The segmentation boundaries between somites are primarily determined by the anterior border of the Tbx6 protein domain in the presomitic mesoderm (PSM). The authors had previously shown reciprocal regulation of Tbx6-Mesp2-Ripply1 in this context. In this manuscript they go on to examine the mechanisms that underlie the critical protein degradation of Tbx6 at this border, with a combination of biochemical approaches and a clever use of an in vitro ES cell system. This is an important and well executed study. However the authors should consider the following point:

The authors show that Ripply2 actually degrades Tbx6 proteins in PSM-like cells induced from ES cells. Although this is one of the highlights of this paper, the finding was predictable from previous work. On the other hand, the authors noticed another unexpected and interesting feature of Tbx6 degradation, namely that Tbx6 degradation does not occur ubiquitously, but is specific for the PSM cells. If the authors could show some additional data to explain a possible mechanism by which this specificity occurs, the impact of the paper would be increased.

---

## [Author Response]

The regular formation of somites, that takes place progressively on the anterior/posterior axis of the embryo, is a fascinating process. The segmentation boundaries between somites are primarily determined by the anterior border of the Tbx6 protein domain in the presomitic mesoderm (PSM). The authors had previously shown reciprocal regulation of Tbx6-Mesp2-Ripply1 in this context. In this manuscript they go on to examine the mechanisms that underlie the critical protein degradation of Tbx6 at this border, with a combination of biochemical approaches and a clever use of an in vitro ES cell system. This is an important and well executed study. However the authors should consider the following point:The authors show that Ripply2 actually degrades Tbx6 proteins in PSM-like cells induced from ES cells. Although this is one of the highlights of this paper, the finding was predictable from previous work. On the other hand, the authors noticed another unexpected and interesting feature of Tbx6 degradation, namely that Tbx6 degradation does not occur ubiquitously, but is specific for the PSM cells. If the authors could show some additional data to explain a possible mechanism by which this specificity occurs, the impact of the paper would be increased.

We understand the reviewer’s comment. We expect that the specificity is due to a tissue-specific E3 ubiquitin ligase. However, we have not yet identified such an E3 ligase responsible for Tbx6 degradation in the PSM. Instead, we presented several data indicating that some PSM-specific factor or mechanism is involved in the Tbx6 degradation.

1) Tbx6 degradation was not induced by Ripply2 in HEK293 cells.

2) Ectopic induction of Ripply2 in the PSM resulted in the down-regulation of T-family proteins, Brachyury and Tbx6 in the PSM (Zhao et al., 2015), but the forced expression of Ripply2 in the heart lineage did not affect T-family proteins, Tbx5 (cardiomyocyte marker) or Tbx18 (epicardium marker) (Figure 1—figure supplement 2).

3) We confirmed that Tbx6 was more ubiquitinated in the PSM-fated ES cells compared with in undifferentiated ES cells.